# Entomopathogenic Fungi: An Eco-Friendly Synthesis of Sustainable Nanoparticles and Their Nanopesticide Properties

**DOI:** 10.3390/microorganisms11061617

**Published:** 2023-06-19

**Authors:** Ritu Bihal, Jameel M. Al-Khayri, A. Najitha Banu, Natasha Kudesia, Farah K. Ahmed, Rudradeb Sarkar, Akshit Arora, Kamel A. Abd-Elsalam

**Affiliations:** 1Department of Zoology, School of Bioengineering and Biosciences, Lovely Professional University, Phagwara 144001, India; 16ritu2000@gmail.com (R.B.); natashakds776@gmail.com (N.K.); rudradebsarkar2001@gmail.com (R.S.); akshitarora12344321@gmail.com (A.A.); 2Department of Agricultural Biotechnology, College of Agriculture and Food Sciences, King Faisal University, Al-Ahsa 31982, Saudi Arabia; 3Biotechnology English Program, Faculty of Agriculture, Cairo University, Giza 12613, Egypt; farah.kamel@std.agr.cu.edu.eg; 4Plant Pathology Research Institute, Agricultural Research Center, Giza 12619, Egypt; kamelabdelsalam@yandex.com

**Keywords:** nanotechnology, pest management, biosynthesized silver nanoparticles, entomopathogenic fungi, toxicity mechanism

## Abstract

The agricultural industry could undergo significant changes due to the revolutionary potential of nanotechnology. Nanotechnology has a broad range of possible applications and advantages, including insect pest management using treatments based on nanoparticle insecticides. Conventional techniques, such as integrated pest management, are inadequate, and using chemical pesticides has negative consequences. As a result, nanotechnology would provide ecologically beneficial and effective alternatives for insect pest control. Considering the remarkable traits they exhibit, silver nanoparticles (AgNPs) are recognized as potential prospects in agriculture. Due to their efficiency and great biocompatibility, the utilization of biologically synthesized nanosilver in insect pest control has significantly increased nowadays. Silver nanoparticles have been produced using a wide range of microbes and plants, which is considered an environmentally friendly method. However, among all, entomopathogenic fungi (EPF) have the most potential to be used in the biosynthesis of silver nanoparticles with a variety of properties. Therefore, in this review, different ways to get rid of agricultural pests have been discussed, with a focus on the importance and growing popularity of biosynthesized nanosilver, especially silver nanoparticles made from fungi that kill insects. Finally, the review highlights the need for further studies so that the efficiency of bio-nanosilver could be tested for field application and the exact mode of action of silver nanoparticles against pests can be elucidated, which will eventually be a boon to the agricultural industry for putting a check on pest populations.

## 1. Introduction

Globally, the agriculture industry is facing a major issue of loss of crop yield due to several abiotic and biotic factors. However, insect pests are causing a major decline in agricultural production. Insects can feed on a wide variety of plants, including crops, weeds, forest trees, and medicinal plants. They also infest food and other stored products in bins, godowns, and packages, causing massive food loss and the deterioration of food quality. Insects that cause more than 10% harm are classified as major pests, whereas those that cause damage between 5 and 10% are classified as minor pests [1]. Globally insect pests are causing an 18–20% loss in crop production, and the annual cost of crop loss is approximately US $470 billion [2].

Therefore, a variety of chemicals, particularly insecticides, are utilized to protect the valuable crops that are harmed by pests. Chlorantraniliprole, cyantraniliprole, novaluron, neonicotinoids, fipronil, farnesyl acetate, emamectin benzoate, phoxim, and pyrethroid are a few examples of pesticides used to control pests. Methoprene, fenoxycarb, and pyriproxyfen are a few examples of juvenile hormone analogues used as insecticides, but all of these pesticides harm the ecosystem, and many pests have acquired resistance to them [3,4,5]. To reduce the damage of pests in fields, many chemical pesticides such as chlorantraniliprole and cyantraniliprole are used, but extreme usage of these chemical pesticides has led to the formation of insect resistance, pest resurgence, and various environmental hazards. The deteriorating activity of chemical pesticides has diverted researchers to focus on eco-friendly control tactics [6].

An entomopathogenic fungus (EPF) is a microorganism with the ability to infect, parasitize, and kill arthropod pests. Generally used as an alternative to conventional chemical pesticides in organic farming, they also have their roots in the fields of biotechnological processing and Chinese medicine. These EPF do not belong to a single monophyletic group but, instead, there are approximately 12 known species from *Oomycetes*, 399 species from *Microsporidia*, 65 species from *Chytridiomycota*, 474 species from *Entomophtoromycota*, 283 species from *Basidiomycota*, and a total of 476 species from *Ascomycota*. Out of all these, the ones belonging to the phylum *Ascomycota* and the phylum *Entomophtoromycota* are found more frequently in nature. The highest entomopathogenic efficacy among them was observed from the ones belonging to the phylum *Entomophtorales* (like *Conidiobolus*, *Furia*, *Erynia*, and *Entomophaga*), but due to a lack of proper breeding technologies in laboratories, they are avoided as components for bio-preparation. However, the saprophytes belonging to the phylum *Ascomycota* are commonly utilized. Other species of entomopathogenic fungi that are utilized as biopesticides are from the genera *Beauveria*, *Paecilomyces*, *Metarhizium*, *Lecanicillium*, and *Isaria*. Since these fungi have an array of applications, they can be utilized to eradicate a wide spectrum of insect pests [7,8,9,10,11,12,13,14].

Entomopathogenic fungi directly penetrate the outer integument of an insect’s body to infect it. The process of infection starts with spore attachment to the exoskeleton of the arthropod due to electrostatic and hydrophobic forces, along with the activation of the lytic enzymes and secondary metabolites [15]. The fungal hyphae begin to develop after invading the insect’s body cavity. Some EPF produces blastoconidium or spores that invade the insect’s hemolymph and develop secondary hyphae that remain in the tissues and impair the physiological processes. The body of the insect, which is primarily soft, stiffens as the infection progresses due to fluid absorption by the pathogenic fungus [16]. However, the effectiveness of EPF is restricted by limitations such as short shelf life, slow action, high application cost, a lack of persistence, and a poor infection rate in harsh environmental conditions. They are frequently slow-acting and necessitate a high application rate and extensive spray coverage. Therefore, one way to enhance the efficacy of these fungi is by using nanoparticles [17].

Nanotechnology is a novel approach to agricultural research that helps to reach the goal of environmentally benign, cost-effective crop protection and production. It provides a way to synthesize nanoparticles (1–100 nm) that may be used as nanopesticides to eradicate various agricultural pests. Nanoparticles are the primary building blocks of nanotechnology. Nanoparticles can be characterized based on properties such as morphology, shape, size, etc. [18]. When nanoparticles enter an insect’s intracellular area after penetrating the exoskeleton, they attach to phosphorus or sulfur in proteins or DNA, which leads to the denaturant of organelles and enzymes. Nanoparticles can reduce membrane permeability, resulting in cellular function loss and cell death. Nanoparticles can be administered judiciously and are swiftly taken up by cells, which slows the emergence of resistance in pests. Nanoparticles can be used in a variety of ways for effective pest management, such as a combination formulation of metal and another material that is virulent against a certain insect [19,20].

Microorganisms provide new sources for creating nanostructures. Microorganism biotemplates are made from nature and have evolved over thousands of years to exhibit distinctive structural and material characteristics. When compared to other micro/nanofabrication methods, they are vastly superior and more affordable for producing complex and heterogeneous structures. In recent years, fungi, a broad group of microorganisms with a variety of species in nature, have also been used to make nanoparticles. Among all biosynthesized nanoparticles, silver nanoparticles have prospective applications in different fields of science, such as life sciences, agriculture, forensic science, and food chemistry. Nowadays, silver nanoparticles are gaining much popularity due to their insecticidal properties. Numerous studies have demonstrated that biologically synthesized silver nanoparticles are quite effective against a range of insect species. Additionally, the use of biosynthesized nanosilver, more specifically silver nanoparticles produced by the entomopathogenic fungus, has grown in favor. It has been confirmed that the minimum concentration of biosynthesized nanosilver leads to 100 per cent mortality in different pest species. EPF is becoming a desirable candidate for the synthesis of nanoparticles because of their high yields, ability to produce large quantities of proteins, ease of handling, and low toxicity of the residues [21,22,23,24,25,26]. In this regard, the main aim of this review is to highlight EPF as an emerging candidate for the synthesis of nanoparticles. The review also spotlights the efficacy of biosynthesized nanoparticles against pest species. Finally, the review concludes that there is a need for further research as very little is known regarding the mode of action of biosynthesized nanoparticles in insect pests.

## 2. What Are Entomopathogenic Fungi?

Generally, the term entomopathogens refers to the organisms that are pathogenic to insect pests. Entomopathogens can be naturally occurring such as certain bacteria, fungi, viruses, and nematodes, which will infect a certain number of insect pest species and will help us to manage crop growth. Entomopathogenic fungi were first among the microorganisms used as biocontrol agents against pests. The first successful use of biocontrol methods to eradicate pests was initiated in 1762 to control the population of sugarcane red locust *Nomadacris septemfasciata* through the introduction of the mynah bird *Acridotheres tristis* from India into Mauritius. However, the use of EPF, *Beauveria bassiana* as a biocontrol agent first occurred in 1956 in China against the sweet potato leaf weevil *Cylas formicarius* in 1959 [27,28,29]. EPF is recognized as a class of bioinsecticides that can be used to eradicate insect pests. They are regarded as effective biological insect control agents. Throughout a significant portion of its life cycle, EPF is mostly harbored in natural or cultivable soils of the terrestrial ecosystem. These organisms are more abundant in soils with more organic matter and acidity [30,31]. They are phylogenetically different, unicellular or multicellular eukaryotic microorganisms that have the potential to irradicate the pest by attacking them, causing infection, using them as hosts, and ultimately killing the pest to keep their population below the economic injury level. They are facultative or obligatory pathogens that have a high survival rate and reproduction capacity. Generally, these fungi are categorized into six major Phylum: Basidiomycota, Ascomycota, Oomycetes, Microsporidia, Entomophtoromycota, and Chytridiomycota [9,32,33,34]. In contrast to all these, phylum Ascomycota contains a wide range of EPF, from less specious orders such as Pleosporales, Myriangiales, and Ascosphaerales to very diverse subgroups within the relatively well-known order Hypocreales [35]. Order Hypocreales consists of the most prominent EPFs such as *Metarhizium*, *Beauveria*, *Torrubiella*, *Aspergillus*, *Paecilomyces*, *Tolypocladium*, *Aschersonia*, *Culicinomyces*, *Lecanicillium*, *Hypocrella*, *Hirsutella*, *Cordyceps*, *Samuelsia*, etc., which can kill many insects [36]. Ophiocordyceps, a genus in the order Hypocreales, infects ants, while *Hypocrella*, a genus found in tropical and subtropical regions, exhibits pathogenesis in scale insects and white flies [37]. The genus *Ascosphaera* within the order Ascosphaerales consists of 30 species of EPF that specifically infect bees. Some fungal species cause chalkbrood disease in bees, while others act as saprophytes and feed on the nest material, honey, cocoon, and larvae of the bee [38].

Entomopathogens have advantages over chemically synthesized insecticides as they are eco-friendly, precise, and safe to use. The term entomopathogen was first coined in a study. It described the microbes that aid in insect population control and maintain them within limits that caused no economic damage to the crop plants [39]. EPF has a larger host range compared to entomopathogenic bacteria and viruses as they are host-specific, hence they can infect both soil-dwelling and above-ground insect pests. Pests directly affect crop production by feeding on crops, therefore reducing the yield and quality of the crops. Even more devastating consequences are faced by farmers if the population of a particular pest is huge as many pests are believed to be found in millions globally. The impact of agricultural pests is so detrimental that, globally, it has resulted in a 40% reduction in the total agricultural output. It is also estimated that almost 10.80% of the agricultural losses worldwide are caused by insect pests only after the green revolution [1,40]. Conventionally, chemical pesticides were a reliable method of insect pest control because they required low effort for application, showed good efficacy, and were expedient. However, in 80% of cases, overuse of chemical pesticides has resulted in the development of resistance to more than one class of insecticide [41,42]. Entomopathogenic fungi are natural insect population-control agents as they are naturally pathogenic to a range of insect pests and they are derived from nature; hence, they show little to no adverse effects on the environment. Different EPFs have different host choices and, hence, can be specialists or can infect a variety of insect hosts making them a generalist pathogen. Other pathogenic organisms such as protozoans, viruses, and bacteria needed to be ingested or enter the host body in some way to cause the infection, but entomopathogenic fungi can utilize the nutrients present in the body cuticle of the insect pest to colonize the host body, hence making them a good entomopathogenic agent to utilize [43].

## 3. Effect of Different Entomopathogenic Fungi against Pest

Entomopathogenic fungi are parasitic microorganisms that are capable of infecting and killing a wide range of insects. In organic farming, these fungi are used as a biocontrol agent and serve as a safer, more effective, and eco-friendly method than chemical insecticides [8,11,13]. EPFs such as *Metarhizium*, *Beauveria*, and *Trichoderma* are parasitic microbes that can infect and kill a variety of arthropods. They directly penetrate the outer integument of the insect’s body to infect and kill them [15].

In tropical countries, one of the major fruit crops is the papaya (*Carica papaya Linn*.). Papaya is well known for its nutritional, therapeutic, and prophylactic values worldwide. Papaya Ring Spot Virus (PRSV) can cause crop loss from 10% up to 100% every year [44]. The vector for the PRSV is the aphid *Aphis gossypii* and they tend to transmit this virus in a very non-persistent manner [45]. The utilization of chemical pesticides is detrimental to the environment, as well as to human health. An experiment was performed to study the efficacy of indigenous EPF, which was collected from the agriculture fields of North 24 Parganas during the pre-monsoon period. From the soil, a range of fungi was isolated including 11 isolates of *Trichoderma*, 9 isolates of *Aspergillus*, and 8 isolates of *Penicillium*. A total of 20 isolates out of the 40 isolates extracted showed entomopathogenic tendencies against the third- and fourth-instar larva of the aphids of papaya. However, only 3 isolates out of these 20 showed proper entomopathogenic activities, and those isolates were identified as *Penicillium* sp. (Nlg 1), *Fusarium* sp. (khr4), and *Beauveria* sp. (deb4). The efficacy test of these fungi on 10 aphids suggested that the efficiency of *B. bassiana* was greater than *P. verrucosum* followed by *F. equiseti.* These entomopathogenic fungi produce spores that germinate on the surface of the cuticle of the aphis, killing as a result of penetration of the cuticle [46].

Whitefly (*Bemisia tabaci*), a hemipteran insect with several genetically distinct species, is one of the most economically important insect pests of ornamental and vegetable plants throughout the world’s tropical and subtropical climates. The major threat to crops is the nymphs and the adult whiteflies as they feed directly on the plants, as well as act as vectors for various plant viruses. Whiteflies produce honeydew while feeding, which reduces the rate of photosynthesis in plants. Chemical pesticides are commonly used to control whitefly populations but they harm the environment and endanger the safety of non-target organisms. It has been discovered that insects are becoming resistant to these chemicals. EPF can be used as an alternative to chemical pesticides because it is derived from the environment and is a natural enemy of whiteflies. The majority of entomopathogenic fungi are found in the Zygomycetes (phylum-Zygomycota), Hyphomycetes (phylum-Deuteromycota), and Laboulbeniales and Pyrenomycetes (phylum-Ascomycota).

Entomopathogenic fungi are also easy to isolate because they can be extracted from cadavers of insects and soil or grown in an artificial medium. Some of the EPF that is commonly produced commercially and have been documented to act as biological control agents are *B. bassiana*, *M. anisopliae*, and *I. fumosorosea*. Some of the genera of EPF that are known to kill the developmental stages of white flies are *Aschersonia*, *Verticillium*, and *Isaria*. *Clonostachys rosea* is another EPF that has been found to have pathogenic effects on the fourth-instar nymph and adult stages of *B. tabaci*. The entomopathogenic fungi *Aschersonia aleyrodis* could be used as a potential biological control agent for *B. tabaci* as it is documented to be able to efficiently parasitize the whiteflies with greater than 50% mortality within 7 days. The survival rates of the *Bemisia tabaci* first, second, and third instars were also significantly reduced after treatment with *A. aleyrodis* in a greenhouse [47].

Throughout Appalachia, one of the foundation tree species is the American Chestnut (*Castanea dentate*), which once numbered 4 billion in the early 1900s. However, ever since the commercial value of chestnut wood increased to nearly USD 750,000 today, chestnut forests saw a huge amount of deforestation. Following deforestation, there was also the introduction of chestnut blight (*Cryphonectria parasitic*), which, coupled with the deforestation, led to the rapid decline in the chestnut population to the point where there was no chestnut tree canopy in the eastern forest. Fortunately, chestnut-blight-resistant trees were introduced into the forest, which was able to seize the spread of chestnut blight. Unfortunately, following this success was the resurgence of another mortal enemy of the chestnut trees known as the (*Curculio sayi*) Lesser Chestnut Weevil. *C. sayi* was a prominent pest of the chestnut trees with an infestation rate between 50 and 75% considered to be normal, while some reached a 100% infestation rate.

This infestation of the lesser chestnut weevil larvae and excreta also decreased the commercial value of the infested tree due to the quality of the wood being adversely affected. The *C. sayi* larvae can directly damage the chestnut wood physically, and even if they are heat-killed post-harvest, they still reduce the nut meat substantially. Moreover, the larvae will facilitate the growth of *Aspergillus* fungus, which will produce the diarrheagenic toxin emodin. As with any other chemical pesticides, the pesticides administered to control the *C. sayi* population will eventually adversely affect the environment, hence a greener and more eco-friendly approach is required to control the lesser chestnut weevil population. Since the adult lesser chestnut weevil will climb the canopy of the chestnut trees during the spring season, utilizing EPF as a biological control agent to control them seems to be a favorable alternative to chemical pesticides. For the experiment, an entomopathogenic fungus called *Beauveria bassiana* was applied to each trunk of mature chestnut trees in a 0.5 m band encircling the circumference of the chestnut trees and terminating approximately 1 m above the ground level. To measure the efficacy of the EPF, pyramid and trunk traps were used to monitor the adult lesser chestnut weevil population. After the application of the EPF on the trunk of the chestnut trees, the catch rate of the *C. sayi* adults in the trap was significantly reduced compared to the control. Furthermore, chestnuts treated with entomopathogenic fungus had a much lower chance of weevil infection, both as the major effect and when compared to the control. The reduction of the trap catch of *C. sayi* was attributed to two possibilities. The first possibility was that the application of the EPF on the trunks of the mature chestnut trees acted as a natural repellent for the adult lesser chestnut weevil to climb the top of the canopy, and secondly, the application of the EPF showed a potentially lethal effect on the adult *C. sayi* as the unfortunate insects that came in contact with the EPF contracted the fungus, which slowly colonized them and ultimately killed them in the process in a matter of days via systematic fungus infections. Another EPF in addition to *B. bassiana* that was able to survive the harsh northern winters and also showed better efficacy in controlling the lesser chestnut weevil population was *S. feltiae* [48].

One of the economically important pests in commercial crops is from the order Hemiptera, which are often referred to as true bugs. The name of the infamous bug is *Elasmolomus pallens*, and it is a post-harvest pest insect that can be found worldwide in different climatic zones including subtropical, tropical, and temperate. The success of the bug is linked to its feeding choice of legume plants such as *Arachis hypogea L*. in Sub-Saharan Africa. A significant chunk of the economic loss of peanuts is caused by the adults and nymphs of *E. pallens* during the harvest and storage of the peanuts. These bugs use their piercing mouth part called the Rostrum for feeding. The aftermath of the *E. pallens* infestation is that the seeds will be shriveled, which will result in increased content of fatty acids and a rancid flavor in the oil made from those seeds. The entomopathogenic studied was *A. flavus*, and the fungus was isolated from the colony of *A. flavus*. The bioassays used in this experiment included a dose–response bioassay, an enzyme bioassay, and a correlation between mortality rates and enzymatic activities. In the dose–response bioassay, the conidial concentration was directly proportional to the mortality rates. With the increase in the conidial concentration, the mortality rates also increased. However, the oil-formulated conidia were more virulent, with 100% combined mortality achieved after 8 days compared to 92% mortality achieved after 9 days of treatment with conidia formulated in the surfactant Tween 80. Both mortality rates were higher compared to the control, which was approximately 10% after 10 days. In the enzyme assay, it was observed that the enzyme activity increased with the number of days of culture, which later decreased. *A. flavus* showed maximum protease activity on the eighth day, chitinase activity on the eighth day, and lipase activity on the sixth day. The correlation between *E. pallens* mortality rates and enzyme activities of cuticle-degrading enzymes such as protease, lipase, and chitinase was found to be positive. As a result, the study confirmed that *A. flavus* fungal isolates were a potential biocontrol entomopathogenic fungal agent for controlling seed bug (*E. pallens*) populations. The efficacy of *Aspergillus flavus* on the mango seed weevil (*Sternochetus mangiferae*) was also investigated, and it was found to have an 80% mortality rate [43].

### 3.1. Secondary Metabolites of Entomopathogenic Fungi and Their Role in Pest Infection

It has been confirmed that entomopathogenic fungi secrete a wide variety of secondary metabolites, which are low-molecular-weight organic compounds. By lowering the target species’ resistance level and ultimately causing nervous system damage, these secondary metabolites aid in the effective infection of pests and other target organisms. Secondary metabolites are classified chemically as amino acid derivatives, peptides, cyclic depsipeptides polyketides, terpenoids, and peptide hybrids [14,49]. It has been stated that the secondary metabolites are synthesized through gene clusters such as non-ribosomal peptide synthetases (NRPSs), hybrid NRPS–PKS genes, etc. The presence of more secondary metabolites in the entomopathogenic fungal genome than in the genome of typical fungi has been demonstrated during the sequencing of fungi. When these secondary metabolites come into touch with insect tissues, they become active. Entomopathogenic fungi gain pathogenicity through secondary metabolites. However, the bulk of these secondary metabolites are dormant and need activators such as host exposure or environmental stress. The generalist insect pathogens *Metarhizium* and *Beauveria* infect more than a thousand pest species. Both of these entomopathogenic fungi include more than 80% distinct secondary metabolite genes according to genomic analyses. *Beauvericin* and *bassianolide*, both generated by *Beauveria* spp., as well as destruxins, which are mostly produced by *Metarhizium*, are the secondary metabolites that are present. These compounds are known to play a significant function in giving entomopathogenic fungus pathogenicity [50,51].

The effect of secondary metabolites secreted by *Beauveria bassiana* on the adult pest *Eurygaster integriceps* was studied in previous research [52]. According to the study, the metabolites inhibited phagocytic activity, which ultimately changed the way the immune system worked. Additionally, it was shown that the number of hemocytes rapidly decreased as the metabolite concentration increased. *Plutella xylostella* is a cosmopolitan pest of economically important crops. The toxic crude metabolites extracted from *Isaria fumosorosea* in Czapek Dox liquid medium were used against the third larvae of *P. xylostella*. The highest mortality of 91 percent was recorded after 6 days of treatment, which showed the maximum mortality. Therefore, it was concluded that the active microbial metabolite compound can be extracted, which could lead to the development of natural products for plant protection [53]. The effects of secondary metabolites from *Isaria fumosorosea*, *Beauveria bassiana*, and *Paecilomyces variotii* on the feeding, growth, fecundity, and hatchability of *Spodoptera litura* were also investigated. The secondary metabolites were extracted through the solvent extraction method. It was observed that the metabolites had an immense impact on the fecundity and hatchability of the pest [54].

The most effective entomopathogenic fungus against insect pests is *Cladosporium cladosporioides*. Tests on cotton aphid nymphs and adults were conducted on the extract of this fungus’ secondary metabolites. At concentrations of 128 ppm and 154 ppm, the *C. cladosporioides* extract was shown to be most efficient against both nymphs and adults. The 3-(4-hydroxy-6-pyranonyl)-5-isopropylpyrrolidin-2-one) and 3-(4-hydroxy-6-pyranonyl)-6-pyranonyl)-pyrrolidin-2-one were also found in the fungal extract after GCMS analysis. The insecticidal property was thought to be caused by both compounds [55]. Different secondary metabolites produced by entomopathogenic fungi and acting as insect growth regulators were examined in a study. The fungi were isolated from the soil to investigate their insecticidal property. The culture was identified as *Lecanicillium attenuatum*, which showed maximum insecticidal activity against the pest *Plutella xylostella.* The fungi strain showed a high level of juvenile hormone antagonists (JHANs) that disrupted the juvenile hormone receptor complex, which led to larval death due to the disruption in insect physiology. Therefore, it was concluded that the entomopathogenic fungi *L. attenuatum* secretes different kinds of secondary metabolites exhibiting JHANs activity [56]. GC-MS analysis revealed various bioactive chemicals, including propanoic acid, ethyl ester, acetic acid, and propyl ester, among others, in the ethyl acetate extract of *Penicillium* sp. for characterization. After 48 h of treatment, *Spodoptera litura* larvae exposed to *Penicillium* sp. ethyl acetate extract demonstrated considerable larvicidal activity, with LC50: 72.205 mg/mL and LC90: 282.783 mg/mL. After 48 h of exposure to the crude extract, high antifeedant activity was seen in 300 g/mL. According to the results of this study, secondary metabolites from *Penicillium* species are useful for controlling *Spodoptera litura* larvae [57].

### 3.2. Mode of Action of Entomopathogenic Fungi against Insect Pest

The mode of action of any entity in the insect body can be through oral ingestion or contact, but entomopathogenic fungi do not need to be ingested to cause infection. Entomopathogenic fungi can directly infect the insect by penetrating the insect’s exoskeleton. The infection process is mediated by several adhesions, secondary metabolites, and lytic enzymes [14]. The initial step of infection is a fungal invasion. The process of fungal invasion starts with the attachment of conidia of insect pathogenic fungi on the insect cuticle, which is the first barrier. The composition of an insect’s cuticle varies depending on its developmental stage. The outermost layer of the insect cuticle is the epicuticle, which is composed of cuticulin (lipoprotein), wax, and some organic compounds, and is followed by the procuticle. The endocuticle, mesocuticle, and exocuticle are the three types of procuticle. The procuticle primarily constitutes chitin and proteins. The exocuticle consists of proteins such as melanin and carotin, which provide hardness, impermeability, and dark color to the chitin. The endocuticle is the third innermost layer of the epidermis that surrounds and protects the internal structures of the insect [58,59,60]. The attachment of spores to the insect’s outer integument layer occurs by first using electrostatic and hydrophobic forces followed by activation of the lytic enzymes, low-molecular-weight proteins, and secondary metabolites [15]. During germination and swelling, a conidium may attach to the cuticle or secrete mucus for adhesion. Some infective sessile spores have an adhesion drop at the spore end to aid in insect attachment [61,62]. *B. bassiana* and *M. anisopliae* both produce hydrophobic conidia with a surface rodlet layer composed of proteins known as hydrophobins, which can easily interact with the hydrophobic outer lipid layer. Mad1 and Mad2 are adhesion genes found in *M. anisopliae*. These adhesion proteins contain signal peptides, threonine-proline-rich regions implicated in adhesion mediating, and putative glycosylphosphatidylinositol anchor sites that would localize the proteins to the plasma membrane. The Mad1 gene may not specifically target an adherence gene, and its loss results in low spore germination and adhesion, resulting in a reduction in blastospore number and virulence. These genes are responsible for passive adhesion and adsorption, whereas the Mad2 gene is a target-specific gene for the active stage. At least two hydrophobins (Hyd1 and Hyd2) are involved in rodlet layer assembly in *B. bassiana*, which contributes to virulence adhesion to hydrophobic surfaces and cell surface hydrophobicity [63]. After adhesion, the cuticle penetration of pre-germinated spores involves some structures and general processes, and the mechanisms of penetration may vary for each fungus. It is required for fungus to penetrate the cuticle to reach the nutrition and energy sources. Cuticle penetration requires the collaboration of mechanical, enzymatic, and physical activities. The fungus enters the cuticle through a germ tube or an appressorium that injects an infection peg into the cuticle [64,65]. Entomopathogenic fungi secrete many cuticle-degrading enzymes, chitinases, esterases, proteases (chymoelastase, collagenase, trypsin, esterase, and chymotrypsin), and lipases. Moreover, the secretion of enzymes endoproteases and aminopeptidase is associated with appressorium formation. *B. bassiana* requires four catalases (catB, catP, catC, and catD), a hydrocarbon carrier protein (Acyl-CoA oxidase), eight cytochromes P450 genes (CYP52-X1, CYP655-C1, CYP5337-A1, CYP52-G11, CYP539-B5, CYP617-N1, CYP53-A26, and CYP584-Q1), and long-chain alcohol (3-oxoacyl carrier protein reductase) for the digestion of lipids in the cuticle [66,67,68,69]. The fungal hyphae begin to develop after invading the arthropod’s body cavity. Some entomopathogenic fungi produce blastoconidium or spores that invade the insect’s hemolymph and develop secondary hyphae that remain in the tissues and impair physiological processes. In the hemocoel of an insect, fungi release various mycotoxins such as *M. anisoplia*, which produce cytochalasins and Destruxins (depsipeptidesdestruxin B and desmethyldestruxin B). *B. bassiana* produces beauvericin, brassinolide, beauverolides, and isarolides [70,71]. The body of an insect, which is primarily soft, stiffens as the infection progresses due to fluid absorption by the pathogenic fungus, and ultimately insects die due to inadequate nutrition, mycotoxin, and tissue impairment. However, entomopathogenic fungi require specific environmental conditions to germinate and cause infection, and their shelf life is limited. Furthermore, preventive treatment is difficult because pests must be present before the pathogen can be usefully applied. They are slow-acting and necessarily involve a relatively high rate of application and extensive spray coverage. Under adverse environmental conditions, there is a lack of persistence and a low rate of infection. Therefore, to enhance its effect on pests, mycosynthesized nanoparticles are being studied [17,49].

## 4. Different Routes of Synthesis of Nanoparticles

Physical and chemical techniques are used in the production of nanoparticles. Both bottom-up and top-down mechanisms can be used to obtain nanoparticles. The top-down approach entails mechanically grinding bulk metals, followed by the stabilization of the resulting nanoscale metal particles with colloidal stabilizers. The generated objects display a broad size variation with a mean diameter of approximately 100 nm. As a result, such a method is typically unsuitable for producing particles with clearly defined geometrical configurations. The bottom-up approach can be regarded as an approach that comes from the other side. There are numerous ways to break down a tiny precursor, which is typically an organometal complex or a salt: Thermal breakdown, optical excitation, which separates the metal atom from the organic residue, or using a reducing agent. Narrow size distribution and particles with a diameter of 1–50 nm are produced by the nucleation of several metal atoms. Therefore, this technique is the most widely employed in NPs chemical synthesis [72,73]. The top-down approach includes physical methods such as laser ablation, evaporation-condensation, and the arc discharge method, and the bottom-up approach includes chemical methods such as microemulsion techniques, chemical reduction, and electrochemical synthetic techniques for the manufacture of NPs. Most chemical and physical methods of producing nanosilver are extremely expensive and involve the use of poisonous, dangerous substances that may cause possible environmental and biological concerns [74,75,76].

### The Greener Synthesis of Metal Nanoparticles

These constraints prompted the recent creation of NPs synthesis based on the concept of green nanotechnology, which was then successfully applied in the biosynthesis of nanoparticles. Extracts from living animals are used, highlighting the usage of amino acids, polysaccharides, vitamins, and proteins/enzymes, as well as other compounds with a coating capacity of metal nanoparticles. The reduction and stabilization operations are carried out in a single step during the biosynthesis process, making it possible to produce nanoparticles quickly [77,78,79,80]. It has been observed that several microorganisms, including bacteria, fungi, *actinomycetes*, and yeast, are capable of synthesizing nanoparticles, mineral crystals, and metallic nanoparticles either intracellularly or extracellularly. Two mechanisms—extracellular and intracellular—are involved, depending on where NPs are synthesized. Extracellular synthesis refers to the generation of NPs outside of the plant or microbial cell as a result of the presence of biomolecules such as enzymes, amino acids, and proteins. Metal ion transition within the cell occurs as a result of an electrostatic interaction between positively charged metal ions and negatively charged cell walls. Metal ions are converted into metal NPs by enzymes found in the cell wall. Since NPs are created inside the cells, intracellular synthesis is a very difficult process. The extracellular technique does away with the extra steps needed for the intracellularly produced NPs to be extracted. This is why the extracellular technique is typically chosen for the creation of nanoparticles [81,82,83].

In this review, the emphasis will be on developments in the synthesis mechanism and elements influencing nanoparticle properties. To comprehend the mechanisms of nanoparticle biosynthesis, microbes are still being used increasingly in recent years. Since bacteria and fungi have more advanced technology than actinomycetes and yeast, their usage in the creation of nanoparticles has attracted greater attention than that of actinomycetes and yeast. Fungi are appealing as reducing and stabilizing agents in the biogenic synthesis of silver nanoparticles due to the production of significant amounts of proteins, high productivity, simplicity in handling, and low toxicity of the residues. Furthermore, the nanoparticles are coated with fungus-derived proteins, which can increase stability and possibly provide biological activity [82,84]. Fungal cultures have advantages over bacterial cultures, such as strong biomass generation and the lack of additional procedures needed to collect the filtrate. Fungi mycelial mass is more resistant to pressure and agitation than plant mycelial mass, making them better suited for large-scale syntheses. Furthermore, it is possible to manipulate fungus metabolism to produce nanoparticles with specific properties, such as size and morphology, by adjusting culture factors such as time, temperature, biomass amount, and pH, among others [85,86]. Figure 1 illustrates different methods for the synthesis of nanoparticles.

## 5. Synthesis of Metal Nanoparticles by Entomopathogenic Fungi

### 5.1. Selenium Nanoparticles

Fungi of the *Trichoderma* genus are widely utilized and studied as a biocontrol agent against pathogenic plant fungi, as well as nematodes and bacteria. The hydrolytic enzymes present in *Trichoderma* tend to degrade the cell wall of pathogens. *Trichoderma* also has secondary metabolites, for example, isoprenoids and peptaibols, which aid in the control of root and foliar pathogens. Studies related to the entomopathogenic effect of *Trichoderma* were scarce as, generally, the fungus interacts with subterranean organisms. However, it is described in certain places that *Trichoderma* species do have entomopathogenic properties, especially against the lepidopteran insects as they are known to contain the chitinase enzyme that degrades the cuticle of the target insect and, when suitable conditions are provided, will negatively affect the peritrophic matrix in silkworms. In addition to biological methods, physical and chemical approaches can be utilized to synthesize the selenium nanoparticles, although these will require the use of a higher quantity of hazardous chemicals, higher thermal conditions, and more acidic pH due to which the biological application of these nanoparticles can become extremely lethal and dangerous. Acknowledging these side effects, the synthesis of selenium nanoparticles via the biological method is preferred as it gives a safer, eco-friendly, and more non-toxic alternative than the conventional way. Additionally, those biologically synthesized SeNPs tend to be more stable as they have a natural coating of organic material on their surface, which will prevent the aggregation of the nanoparticles over a long time [87,88,89,90,91,92,93,94,95].

A polyphagous insect that is responsible for the widespread destruction of crops such as cauliflower, tea, rice, cotton, chili, groundnut, cabbage, capsicum, maize, and tobacco is *Spodoptera litura*. Conventionally, to control the *S. litura* population, chemical pesticides were used; however, they adversely affected human health along with the environment and non-target beneficial arthropods. A study was performed to determine the efficacy of mycosynthesized SeNP from *Trichoderma* and its effect on *S. litura* larva. The synthesis of the nanoparticles was performed utilizing the *Trichoderma* extract, which was added dropwise into 20 mM 100 mL of a selenium solution, which was placed in a magnetic stirrer, and then Na_2_SO_3_ was added to the solution to maintain the pH at 8–12. The reaction mixture was kept in the dark for reduction and the temperature was maintained at 27 °C ± 2 °C. The mixture was also kept in the orbital shaker for 24 h at 120 rpm and any color change was noted. After this, the synthesized nanoparticles were stored in dark conditions until a brick-red-colored sedimentation was observed at the bottom of the conical flask. Following this, they were kept for 3–4 days, after which they were subjected to centrifugation at 8000 rpm, and then they collected the pellet, dried them at room temperature, and stored them for further characterization.

The characterization results obtained indicated that the *Trichoderma* mycosynthesized SeNPs have a 259 nm UV-Vis peak range. FTIR analysis also indicated the presence of many functional groups such as phosphines, anhydrides, nitro groups, alcohols, alkanes, and sulfonates. The morphology of the SeNP was confirmed to be spherical, while the EDaX analysis showed that 87.98% of the Selenium elemental compound was present. XRD confirmed the crystalline structure of selenium with an average size of 137 nm. DLS indicated the SeNP ranged between 40 and 100 nm in size [95,96,97,98,99].

### 5.2. Copper Nanoparticles

Copper metal has long been known for its antimicrobial properties. Copper-based compounds are used in agriculture in a variety of ways, and by reducing these compounds to the nanoscale, their efficacy can be increased by providing more surface area for interaction and enhancing biocidal properties such as DNA damage, oxidative stress, protein and enzyme inactivation, and ROS production [100,101].

The synthesis of copper nanoparticles using an aqueous extract of *Metarhizium robertsii* has been demonstrated against a variety of targeted and non-targeted organisms (*Aedes aegypti*, *Anopheles stephensi*, *Culex quinquefasciatus*, and *Tenebrio monitor* and *Eudrilus eugeniae*, *Eudrilus andrei*, *Artemia nauplii*, and *Artemia salina*, respectively). For the synthesis, 85 mL of 1 mM Copper sulfate was added to 15 mL of an aqueous *Metarhizium robertsii* extract then was carefully heated at 60 °C by placing it on a hot-plate magnetic stirrer. After this, the solution was incubated in the dark for 72 h at 28 ± 2 °C, and the color changed to a dark brown, which indicates the synthesis of CuNPs. The solution was then filtered using Whatman no. 1 filter paper and washed with double-distilled water before being centrifuged at 11,000 rpm for 8 min. The washing methods were repeated multiple times until the unwanted particles were removed. Finally, the pellet was allowed to dry for two days at room temperature before being used in all tests. A UV-Vis spectrophotometer, X-ray diffraction (XRD), Fourier Transform Infrared Spectroscopy (FTIR),-Energy Dispersive X-Ray Analysis (EDaX), an Atomic Force Microscope (AFM), Dynamic Light Scattering (DLS) and Zeta Potential, and a High-Resolution Scanning Electron Microscope (HR-SEM) were used to characterize the CuNPs. In UV-Vis spectrophotometer analysis, CuNPs revealed a significant absorption peak at 670 nm. The XRD analysis revealed three distinct peaks at 29.4389, 35.1523, and 36.8562, suggesting the purity and crystalline nature of CuNPs. FTIR study revealed three functional groups (ArO-H H, -CH2-, and Ar-CH = CHR). The existence of CuNPs was confirmed by EDaX analysis, which produced a peak ranging from 99.8%. The shape of nanoparticles is confirmed as spherical and symmetrical via AFM analysis. The size of the NPs was confirmed by HR-SEM analysis to be 15.67–62.56 nm. DLS analysis revealed that the synthesized CuNPs are polydispersed particles ranging in size from 100 nm to 500 nm. CuNPs have a zeta potential of 20.3 mV, showing their stability. The green-synthesized CuNps showed high larvicidal activity against the larvae of *Aedes aegypti*, *Anopheles stephensi*, *Culex quinquefasciatus*, and *Tenebrio molitor*, establishing CuNPs as a promising candidate for further research as a nanopesticide [102].

### 5.3. Zinc Nanoparticles

The biocompatible nature and the unique properties of zinc nanoparticles make them the best candidate for a variety of applications including biomedical, anti-microbial, etc. However, the chemicals used in the conventional synthesis procedures render this ability of zinc nanoparticles ineffective. To combat this issue in recent times, green synthesis methods for zinc nanoparticles are being developed worldwide and are gaining their deserved spotlight with the scientific community. Green synthesis includes synthesizing nanoparticles, may it be zinc, silver, titanium, silica, or any other, using biological entities such as plants (photosynthesis) or fungi (mycosynthesis). Fungi can be found in a variety of locations, but the most common approach to collecting fungi for mycosynthesis is from the soil or insect cadavers [103]. In a previous study [104], zinc nanoparticles were synthesized using mushrooms of the *Daedalea* species. In the study, 10 g of the mushroom culture was added to 100 mL of distilled water to prepare the extract, which was then slowly mixed in 100 mL of the zinc solution. The zinc solution was prepared by dissolving 1.834 g of zinc acetate dehydrate in 100 mL of distilled water. This solution was held on the hot plate for 6 h at 100 degrees Celsius, with the magnetic stirrer set to 250 revolutions per minute. The color of the zinc oxide solution was conserved, and the pH was determined to be 1.9. That sample was then subjected to centrifugation for 15 min at 15,000 rpm. The pellet was collected while the supernatant was discarded. The nanoparticles were then washed and transferred to a petri dish, which was heated on a hot plate at 300 °C. The resulting dry particles were crushed to a fine powder and kept at room temperature until further usage. The UV Vis spectrophotometric analysis showed the range of wavelength to be between 360 nm and 380 nm at room temperature. XRD analysis was used to determine the crystallinity of the nanoparticles. The structural characteristics of the ZnO nanoparticles were determined using the XRD spectrum. The crystals were normal atom arrays. The XRD patterns validated the phase identity, purity, and crystallite size. The spectra of ZnO NPs show distinct diffraction, with peak values of (100), (002), (101), (102), (110), (103), (200), (112), (201), and (202), respectively. They have a hexagonal crystalline structure. The zinc nanoparticles had an average crystal size of 14.53 nm. The SEM study showed that the nanoparticle’s shapes were irregular and that they were aggregated and agglomerated, which might be credited to the use of extract made from wild sticky mushrooms that acted as a stabilizing and capping agent. The EDX analysis indicated that in the mycosynthesized nanoparticles, the amount of zinc was 58.34% and oxygen was 13.28%. The mycosynthesis of zinc nanoparticles utilized species of *Trichoderma*, namely, *T. reesei*, *T. harzianum*, and *T. reesei*, and the co-culture of *Trichoderma* spp. 1 g of zinc nitrate hexahydrate was dissolved in 10 mL of double-distilled water. Then 2 milliliters of the fungal extract of *T. reesei*, *T. harzianum*, and *T. reesei*, and the co-culture of *Trichoderma* spp. was added to the solution and stirred with a magnetic stirrer for approximately 5–10 min. Following this, the mixture thus obtained was heated in a preheated muffle furnace that was maintained at 400 ± 10 °C in which the mixture boiled, frothed, and the heat-forming foam dehydrated within 3 min. The sample obtained was calcinated at 700 °C for 2 h and the final product thus obtained was stored until further use. When the UV-visible spectrophotometric analysis was performed, the absorption spectrum was observed within 372–374 nm. The FTIR analysis spectrum was observed between 400 cm^−1^ and 600 cm^−1^. EDX analysis showed that the proportion of zinc was 50.36% and that of oxygen was 49.64%. TEM images indicated the presence of small zinc nanoparticles, which were agglomerated [105].

### 5.4. Biogenic Synthesis of Silver Nanoparticles Mediated by Fungi

Fungi’s ability to produce a diverse range of compounds useful in a variety of applications is very promising. Ascomycetes and imperfect fungi, which are microscopic filamentous fungi, as well as other fungi species, are known to produce approximately 6400 bioactive compounds. Due to their tolerance for heavy metals and ability to internalize and bioaccumulate metals, these organisms are frequently utilized as reducing and stabilizing agents. Additionally, the fungus may be readily grown on an industrial scale to manufacture nanoparticles with precise size and morphology (“nano factories”) [80,106]. *Beauveria* and *Metarhizium*, two genera of the entomopathogenic fungus, are frequently isolated from soil. They offer benefits such as strong enzymatic activity, secondary metabolite synthesis, and insecticidal qualities. Consequently, EPF is viewed as the best substitute for chemical and physical synthesis [14,107].

To date, different fungi have been found to synthesize silver nanoparticles. Silver nanoparticles were synthesized extracellularly in a study using *Beauveria bassaina*. Freshly inoculated *B. bassiana* pure culture was placed in an Erlenmeyer flask on a liquid medium that contained KH_2_PO_4_, K_2_HPO_4_, MgSO_4_, H_2_O, (NH_4_)_2_SO_4_, yeast extract, and glucose. For 72 h, the flask containing the medium was incubated in an orbital shaker at 150 rpm at 25 ± 2 °C. The biomass was harvested after 72 h of growth by sieving through Whatman No. 1 filter paper, followed by an extensive wash with distilled water to remove any medium components from the biomass. Then 200 mL of a milli-Q water solution and 20 g of fresh, clean biomass were placed in an Erlenmeyer flask, which was then incubated at 25 °C for 72 h while being stirred in the manner previously described. After incubation, the cell filtrates were obtained by filtering them through Whatman No. 1 filter paper. A 250 mL Erlenmeyer flask was filled with cell filtrate (50 mL), and the final concentration of 1 mM AgNO_3_ (0.017 g/100 mL) was added. The flasks were stirred and incubated in the same manner as previously stated for 120 h at 25 °C in the dark. The nanosilver solution changed color to a brownish-yellow tint and was stored at room temperature in screw-capped vials [108]. Similarly, *Metarhizium* spores were used to create silver nanoparticles. *Metarhizium anisopliae* spores were collected by washing with sterile distilled water and gentle brushing with a paintbrush without affecting mycelia growth. After that, sterile filters were used to filter the spore suspension. Next, a 105 conidia/mL concentration was adjusted as a standard inoculum. In the studies, 105 spores/mL of the medium were inoculated at 32 °C and agitated at 120 rpm for 72 h in 250 mL Erlenmeyer flasks containing 100 mL of potato dextrose broth. The fungal biomass was cleaned before being transferred to a 250 mL Erlenmeyer flask with 100 mL of double-distilled water and incubated for 72 h in an orbital shaker at 100 rpm (28 ± 4 °C). Following incubation, Whatman No. 1 filter paper was used to separate the fungal filtrate. One hundred milliliters of fungal extract were mixed with 1 mM silver nitrate solution (0.16987 mg/mL, 1 mM) and kept in an orbital shaker at 100 rpm (28 ± 4 °C). A brownish-yellow solution indicated the presence of silver nanoparticles [109]. The synthesis of nanosilver from entomopathogenic fungi is shown in Figure 2.

Silver nanoparticles have also been synthesized using *Trichoderma longibrachiatum*. The optimum conditions were a 40 °C temperature, a 24 h time period, pH 12, 3 g of biomass, 4 millimolar of AgNO_3_, and 150 rev/min, for the mycosynthesis AgNPs, which were subsequently purified by lyophilization. These ideal circumstances produced 3 g of AgNPs, which were then subjected to characterization techniques. The average size of the synthesized AgNPs and their respective zeta potential values of 1775 nm and 268 mV were discovered using dynamic light scattering (DLS), which demonstrated their stability. According to the X-ray diffraction (XRD) pattern, the mycosynthesized AgNPs with an average size of 61 nm were crystalline. The field emission scanning electron microscope (FESEM) and high-resolution transmission electron microscope (HRTEM) revealed non-agglomerated cuboidal, spherical, and triangular AgNPs with sizes ranging from 5 to 11.05 nm (HRTEM). Fourier transform infrared spectroscopy (FTIR) confirmed the presence of mycelial cell-free filtrate as a reducing and capping agent [110]. *Penicillium oxalicum*, a fungal endophyte found on the leaf of the *Amoora rohituka* plant, has also been utilized for the biofabrication of nanosilver. After the synthesis, the surface plasmon resonance of AgNPs and the reduction of silver salt caused sharp UV-visible spectra to arise at 420 nm. The presence of functional groups in *P. oxalicum*, which were responsible for the reduction of silver salt into silver nanoparticles, was confirmed by FTIR analysis. Through XRD examination, a high level of crystallinity was discovered, and microscopy-based characterizations such as AFM, TEM, and FESEM indicated evenly distributed, spherically formed nanoparticles [111]. Table 1 depicts different entomopathogenic fungi employed for the synthesis of nanosilver.

## 6. Effects of Mycosynthesized Silver Nanoparticles: A Greener Approach in Controlling Pest Species

Nowadays entomopathogenic-based silver nanoparticles can be used as biopesticides to kill pests. AgNPs are frequently used in the fields of medicine and agri-food due to their versatile nature. AgNPs synthesized by mycogenic processes are used to control pest population as it has the potential to be antifeedant, larvicidal, and environmentally friendly. NPs treatment slowed insect growth and prolonged the larval period by up to 4 days compared to larvae fed with a fungus cell-free extract and untreated castor leaves. *Trichoderma harzianum*, an entomopathogenic fungus, produced Ag nanoparticles that were reported to show the highest mortality rates against *Aedes aegypti*, which is a vector of dengue, with 92% and 96% for the first and second instars, and 100% for the third and fourth instars, as well as pupa, respectively [123]. The effect of silver nanoparticles produced by the insect pathogenic fungus *Beauveria bassiana* on mustard aphids was investigated in a study. The bioefficacy of 25 *B. bassiana* isolates against mustard aphids was tested. The isolates were divided into two groups based on the mortality rate of the mustard aphid: Group I (0–50%; 20 isolates) and Group II (>50%; 20 isolates) (51–100 percent; 5 isolates). This study, which is possibly the first report on the synthesis of nanosilver using the entomopathogenic fungus *B. bassiana* and its effectiveness against (*L. erysimi*) mustard aphid, suggests the potential application of silver nanoparticles for insect management in agriculture [116].

At a 100 ppm concentration, nanoparticles demonstrated 100% antifeedant action and pupicidal activity against first-, second-, and third-instar larvae in *Helicoverpa armigera* treated with *Trichoderma viride*-mediated nanoparticles. After exposure to nanoparticles, glutathione-S-transferase activity increased, and the glucosidase enzyme level was reduced in third-instar larval *H. armigera* [124]. Silver-nanoparticle-loaded fungal metabolites nanoconjugate exhibit high mortality against *Spodoptera litura*. The nanoconjugate shows 100%, 100%, 91.2%,84.3%, 78.4%, and 71.3% mortality against first, second, third, fourth, fifth, and sixth instar larvae, respectively [125]. Nanoparticles produced by *Metarhizium anisopliae* also have insecticidal properties. Against *G. mellonella*, nanoparticles displayed a 22 percent larvicidal activity, but a synergistic interaction between the nanoparticles and gamma-irradiated *M. anisopliae *(which had greater amylase, nitrate reductase, protease, and lipase activities) resulted in a larval mortality rate of 82% [126]. Figure 3 describes that metal nanoparticles such as Ag, Se, Fe, etc., can be synthesized using entomopathogenic fungi such as *Bassiana*, *Fusarium*, *Trichoderma*, and *Isaria.* It has been reported through the above studies that these nanoparticles are highly effective against different pests. Studies have also concluded that out of all metals, Ag in its nanoform has the highest efficacy at the lowest concentration in eradicating insect pests.

Another method for producing nanoparticles with insecticidal activity using these microorganisms is to use fungal extracellular enzymes to synthesize silver nanoparticles (AgNPs). In a previous study, entomopathogenic fungal isolates were used to assess biomass production and the ability to synthesize silver nanoparticles. The study used sixteen isolates of entomopathogenic fungus. An insect model, *Tenebrio molitor*, was used to test the pathogenicity and virulence of the fungi at a concentration of 5 × 106 spores/mL. The study confirmed that *B. bassiana*, *M. anisopliae*, and *I. fumosorosea* isolates are effective anti-*T. molitor* candidates [128]. Lastly, using *Fusarium pallidoroseum* biomass, mycogenic AgNPs were synthesized and tested for efficacy against third-instar white grubs (*Holotrichia* sp), a significant sugarcane pest in western Uttar Pradesh (India). AgNPs were utilized to treat third-instar white grub larvae in vitro, and the lethal dosage (LD50) was calculated using Probit analysis. The chi-square test was used to further evaluate the results and determined that the LD50 was significant at 0.05 levels. The purpose of the study was to demonstrate how AgNPs have a larvicidal effect on white grubs. This study strongly supports the effectiveness of AgNPs as white grub control agents, which may encourage the future replacement of dangerous chemical pesticides [129]. The fungal-mediated SeNPs showed high larvicidal efficacy against the larva of *S. lituria.* The mortality percentage was found to be directly proportional to the concentration of the selenium nanoparticles starting from 25–100 ppm. A remarkably high mortality of 78.49% was observed at a concentration of 100 ppm against the larvae of *S. lituria* compared to a diet treated with the fungal extract, which only gave moderate mortality of 45.72% and 51.05% at the concentration of 100 ppm. Additionally, during the period of larvicidal activity testing, SeNP interference affects the growth of the larva, with them progressively becoming smaller in size, along with a variety of anomalies being observed. The treatment also rendered the larva unable to reach the further stages of growth development of their lifecycle [99]. Table 2 explains the efficacy of entomopathogenic fungi against a wide variety of pest species.

### Mode of Action of Mycosynthesized Nanoparticles

Nanoparticles are nano-sized metal oxides ranging from 1 nm to 100 nm. These particles are well known for their insecticidal properties because of their shape, size, greater strength, depth qualities, optical properties, high electrical conductivity, and high concoction reactivity [134]. However, even after learning about their effective insecticidal properties, information on their mode of entry, penetration, and action against insects is restricted. Few research papers related to cytotoxicity and genotoxicity have mentioned that the uptake of nanoparticles or coated nanoparticles is size dependent. According to their size, nanoparticles follow different pathways to enter the cell. Smaller particles are taken up via the endocytic pathway, large particles with a size of 500 nm are processed by the phagocytic pathway, and large aggregates are taken up via micropinocytosis [135]. To protect crops from pests, many eco-friendly control methods and biocontrol agents are being employed in agriculture, but the efficacy rate of these biocontrol agents (bacteria, fungus, plant extract, and virus) is low. To enhance their effectiveness, these biocontrol agents are being combined with nanoparticles by implementing nano-encapsulation efficiently [20]. Whether it is a metal ion or active bio ingredient that is carried by nanoparticles that are causing penetration and toxicity is still not known. The exposed biosynthesized nanoparticle can enter insects through contact, inhalation, or the oral route [136]. It has been mentioned that nanoparticles can penetrate insects through the skin’s outer layer and move through spiracles, mouth openings, anal prologs, abdominal prolegs, and setae [137]. When nanoparticles enter the cuticles, they adversely affect the physiology and morphology of insects. Nanostructured alumina sorb the wax layer via the surface effect and use triboelectric forces, leading to cuticle dehydration [138]. A *P. anisum* essential-oil-based nanoemulsion showed pigmentation changes in the cuticle, muscular destruction, cellular detritus, thickness, and necrosis in the epidermis, and lost the distinction between the exocuticle and endocuticle in *T. castaneum* [139]. It was also demonstrated that nanosilver causes melanin cuticular pigment loss and the downregulation of Cu-dependent enzymes (tyrosinase and Cu-Zn superoxide dismutase) and that nanosilver combined with membrane-bound Cu transporter proteins causes Cu sequestration, simulating the starvation of Cu and vertical flight ability reduction in *D. melanogaster* [140]. Carbon-silver nanohybrids bind to phosphorus and/or sulfur in biological structures such as DNA and proteins leading to blackened heads and disrupted cells, cuticle membrane organization, and guts in *Culex quinquefasciatus* and *Anopheles stephensi* [141]. It was also observed that when *Chironomus riparius* is exposed to commercial nanosilver, it causes abnormality in the expression level of glutathione S-transferase (GST) genes, which is related to oxidative stress and nanosilver toxicity, which also resulted in the downregulation of CrL15, a ribosomal protein gene, which controls the arrangement of ribosomes and hence protein formation. Moreover, CrGnRH1, a gonadotrophin-releasing hormone gene, and CrBR2.2, a Balbiani ring protein gene, were upregulated, indicating the initiation of an insect’s defense mechanism against silver nanoparticles, and the gonadotrophin-releasing hormone induced signal transduction pathways and reproductive failure [142]. AgNPs ingestion in early larval stages has been linked to impaired crawling and climbing abilities in later larval and adult stages. Increased AgNPs oral dosage during the larval stage causes metabolic changes in protein, carbohydrate, and lipid levels, as well as a reduction in the presence of lipid droplets and an increase in ROS in the larval tissue [143]. AgNPs increase oxidative stress in *D. melanogaster* midgut cells and increase the expression of hsp70 and hsp22, which are proteins found in the endoplasmic reticulum and mitochondria, respectively, and are also in charge of activating caspases, which regulate programmed cell death, membrane destabilization, and potential mitochondrial membrane loss [144]. Figure 4 illustrates the mode of entry of EPF-based nanoparticles through spiracles and its toxicity in insects.

## 7. Conclusions

Agriculture pests can substantially reduce crop yield and quality. Chemical pesticides have been heavily applied as the main strategy for pest control since the 1960s. Although pesticides have been effective at reducing pest populations, it is generally known that their use can have adverse impacts on the environment and the crops themselves. For sustainable crop production, eco-friendly management is therefore required. As they have been proven to be successful in pest management and the production of sustainable agricultural goods, biopesticides can be used as an alternative to chemical pesticides. However, up to now, their efficiency has been constrained by technical difficulties. This review article emphasizes the importance of entomopathogenic fungi in controlling the pest population. In this case, an emerging topic of study called effective nano pesticides for plant protection and pest control offers fresh approaches to designing active chemicals at the nanoscale. Previous studies confirmed that metal nanoparticles are effective against a wide variety of insects and pests. Out of all the metal nanoparticles, silver nanoparticles have been considered an excellent insecticidal agent. Nowadays biologically synthesized silver nanoparticles are gaining a lot of popularity. Biosynthesized nanoparticles are an ecologically safe method of managing insect pest populations. At very low concentrations, entomopathogenic fungi-based nanosilver is particularly efficient against a wide range of pest species. However, not much research has been conducted on this topic as yet. The major goal of this review is to emphasize the rapidly developing field of nanotechnology by emphasizing its potential to offer environmentally friendly and effective options for the management of insect pests in agriculture without endangering the environment. The traditional methods of controlling insect pests, the drawbacks of chemical pesticides, and the potential of EPF-nanosilver as a novel candidate for pest control were also covered in this paper. Finally, the review necessitates the need for further studies so that the exact mode of action of EPF nanosilver can be studied in detail, which would also help in their field application.

## Figures and Tables

**Figure 1 microorganisms-11-01617-f001:**
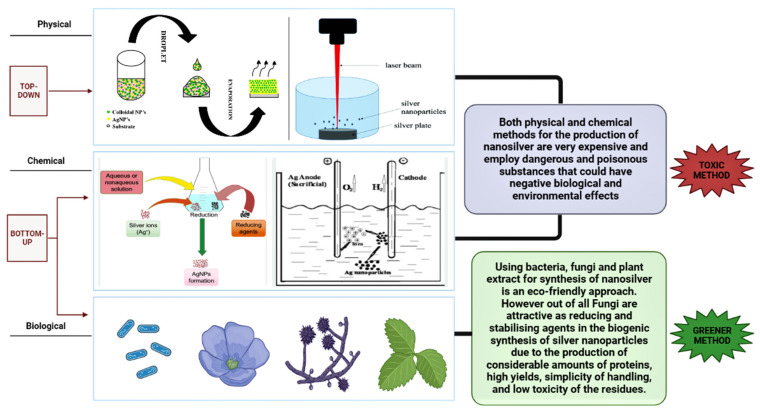
The top-down technique, in which the bulk substance shrinks in size to generate NPs, and the bottom-up method, in which atoms and molecules self-assemble to form nanoparticles, are the two methods for producing nanoparticles. The top-down strategy includes physical methods such as arc discharge, evaporation-condensation, and laser ablation, whereas the bottom-up approach includes chemical methods such as chemical reduction, micro-emulsion techniques, and electrochemical synthetic techniques. Physical and chemical techniques, despite their popularity, are limited in their application due to limitations such as high costs, toxicity, complexity, and age. NPs are created by a variety of biological processes, including fungi (microbes), bacteria, and plants. The biological process is inexpensive, produces a bulk amount, and is eco-friendly.

**Figure 2 microorganisms-11-01617-f002:**
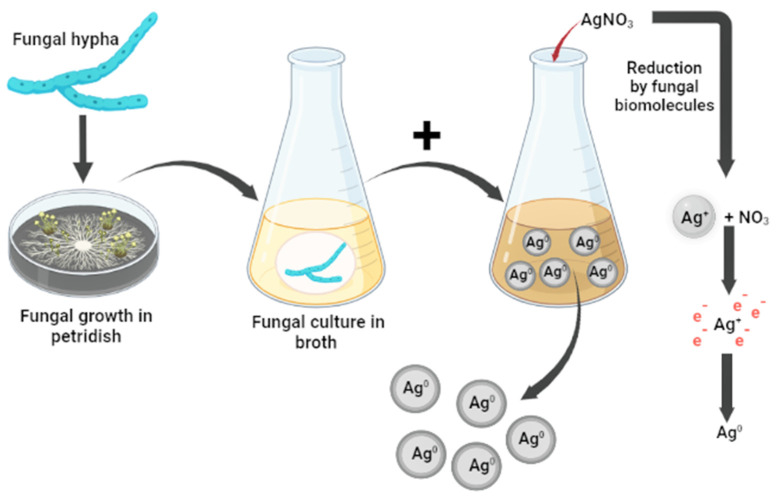
Mechanism for the synthesis of fungi-mediated silver nanoparticles [25].

**Figure 3 microorganisms-11-01617-f003:**
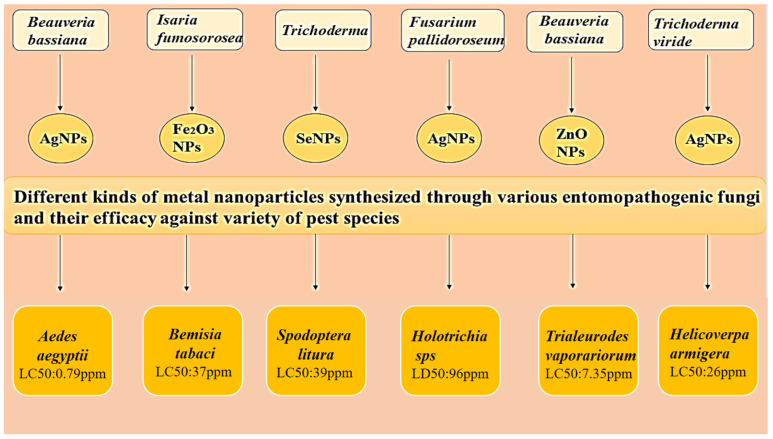
A composite figure providing a brief outline of various metal nanoparticles synthesized using entomopathogenic fungi. These different metal nanoparticles have potential role in controlling variety of pest species. *Beauveria bassiana* based nanosilver has very high efficacy against *Aedes larvae* [106], while *Isaria fumosorosea* synthesized iron nanoparticles are effective against *Bemisia tabaci* [127]. Selenium nanoparticles synthesized using *Trichoderma* has been found to have efficacy against *Spodoptera litura* [98]. Silver nanoparticles synthesized from *Fusarium pallidoroseum* and *Trichoderma viride* is found to be effective in controlling *Holotrichia* and *Helicoverpa sps* [122,126]. Lastly entomopathogenic fungi *Beauveria bassiana* based zinc nanoparticles exhibit high efficacy against *Trialeurodes vaporariorum* [127].

**Figure 4 microorganisms-11-01617-f004:**
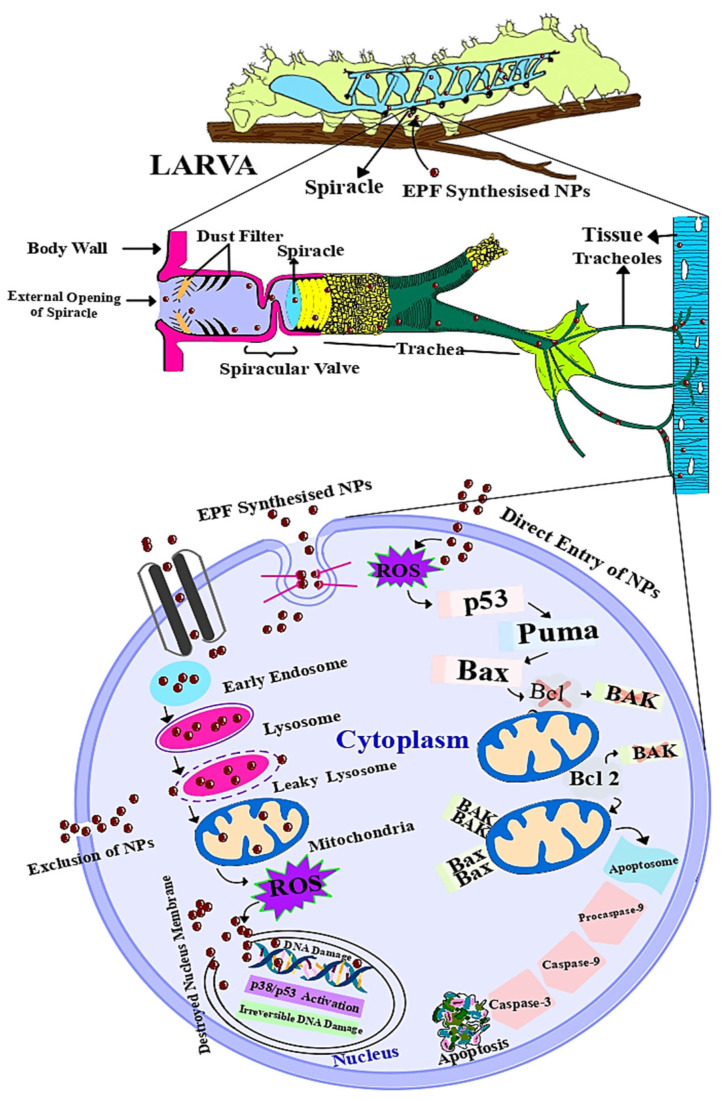
Nanoparticles are very small-sized particles that can easily enter exposed spiracles on the insect body and travel through the trachea and tracheoles into various tissues and cells of the insect, causing cell toxicity and eventually leading to apoptosis and tissue damage. Direct penetration, endocytosis, and cation channels are all ways for nanoparticles to enter the cell membrane. When nanoparticles travel through the endocytic pathway, they are engulfed by lysosomes, causing membrane disruption and exposing lysosomal enzymes containing NPs to other cell organelles such as mitochondria, nuclei, and even neighboring cells. NPs in the nucleus cause irreversible DNA damage. In mitochondria, NPs cause the generation of ROS. The release of excessive ROS induces the apoptosis pathway by increasing the expression of cytochrome c, MDA levels, the disruption of bax/bcl-2 expression, and the cleavage of caspase 3. As a result, cell and tissue death occurs, impairing all physiological and biological processes.

**Table 1 microorganisms-11-01617-t001:** Different entomopathogenic fungi used for the biosynthesis of silver nanoparticles.

Entomopathogenic Fungi Used	Particle Size	Type of Synthesis	Time Taken for Synthesis	Reference
*Aspergillus tubingensis*	35 nm	Extracellular	96 h	[112]
*Penicillium citrinum*	109 nm	Extracellular	24 h	[113]
*Aspergillus foetidus*	20 nm	Extracellular	24 h	[114]
*Isaria fumosorosea*	51 nm	Extracellular	72 h	[115]
*Metarhizium anisopliae*	28 nm	Extracellular	72 h	[109]
*Beauveria bassaina*	20 nm	Extracellular	120 h	[116]
*Trichoderma longibrachiatum*	24 nm	Extracellular	72 h	[117]
*Penicillium oxalicum*	150 nm	Extracellular	96 h	[118]
*Fusarium oxysporum*	25 nm	Extracellular	48 h	[119]
*Fusarium oxysporum*	5 nm	Extracellular	72 h	[120]
*Coriolus versicolor*	444 nm	Intracellular	96 h	[121]
*Aspergillus fumigatus*	25 nm	Extracellular		[122]

**Table 2 microorganisms-11-01617-t002:** Mortality values of entomopathogenic-based nanoparticles against their targeted pest.

Entomopathogenic Fungi	Metal Nanoparticles	Targeted Pest	LC50	Time Taken	Reference
*Penecillium verucosum*	Ag	*Culex quinquefasciatus*	4.91, 5.16, 5.95, 7.83 ppm	24 h	[130]
*Cochliobolus lunatus*	Ag	*Aedes aegypti*	1.29, 1.48, 1.58 ppm	24 h	[131]
*Beauveria bassiana*	ZnO	*Trialeurodes* *vaporariorum*	7.35 ppm	240 h	[127]
*Isaria fumosorosea*	Fe	*Bemisia tabaci*	19.17, 26.10, 37.71 ppm	75 h	[132]
*Beauveria brongniartii*	Fe	*Spodoptera litura*	59 ppm	70 h	[133]

## Data Availability

Data sharing not applicable.

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
