# Peer review of "Entomopathogenic Fungi: An Eco-Friendly Synthesis of Sustainable Nanoparticles and Their Nanopesticide Properties"

_microorganisms, 2023, doi:10.3390/microorganisms11061617_

Round 1
Reviewer 1 Report
In this manuscript, the authors reviewed biosynthesis method of nanoparticles based on entomopathogenic fungi, and also their nanopesticide performance. This study seems to be useful for this field. However, the following problems should be addressed before further consideration of publication:
1. The abstract is long which can be revised to better show the innovation and content of this review. The title needs to be brief and is suggested to be revised with typical keywords such as “biosynthesis”. I understand that “EPF” stand for “entomopathogenic fungi,” yet the abbreviation should be explained in the manuscript for the first time.
2. The Introduction is suggested to be concise to come straight to the point, and several separated paragraphs are suggested. Besides, the advantages of biosynthesis as well as biotemplated fabrication should be briefly revised. The related advances in this field should be added including: 10.1002/sstr.202200356; 10.1021/acsami.1c16859.
3. All the figures need to be revised with higher resolutions and also should be created in consistent layout and style to improve the readability. Text in Figures are often too small to see.
4. The Tables should be checked for consistent formats and expression errors, including units, spaces between values and units, etc.
5. In the subsections, the authors are suggested to add some composite figures of typical research examples contained in this review, which can be helpful for better demonstration.
6. The references should be checked with formats, and many errors exist such as ref. 103, 113, 123, etc. The manuscript should be checked and edited thoroughly for English. For example, expression errors of chemical formula in line 572, and English tense, etc.
7. The structures of this review needs to be improved to be clear considering that seven sections are contained. The authors discuss biosynthesis of various metal NPs yet the related contents are brief and limited.
Author Response
Manuscript Title : Entomopathogenic Fungi: An eco-friendly synthesis of sustainable nanoparticles and their nanopesticide properties
Reference No.: microorganisms-2410859
Reviewers comments
Reviewer # 1
- The abstract is long which can be revised to better show the innovation and content of this review. The title needs to be brief and is suggested to be revised with typical keywords such as “biosynthesis”. I understand that “EPF” stand for “entomopathogenic fungi,” yet the abbreviation should be explained in the manuscript for the first time.
We are very grateful for this suggestion made by the reviewer. The title of the review was decided by all the authors. The word biosynthesis has not been used because already many articles have been published using this keyword. As each and every line in the abstract has a very important meaning it could not be shortened further. Still keeping in mind the suggestion given little modification has been done. Abbreviation has been explained in the manuscript for the first time.
- The Introduction is suggested to be concise to come straight to the point, and several separated paragraphs are suggested. Besides, the advantages of biosynthesis as well as biotemplated fabrication should be briefly revised. The related advances in this field should be added including: 10.1002/sstr.202200356; 10.1021/acsami.1c16859.
We are thankful to the reviewer for this suggestion. As per the comment changes have been made in the introduction. The advantages of biosynthesis has been revised. However we are very apologetic for not including thes studies for biotemplated fabrication as it was not related to our review work.
- All the figures need to be revised with higher resolutions and also should be created in consistent layout and style to improve the readability. Text in Figures are often too small to see.
We are very much obliged for this valuable suggestion given by the reviewer. Resolution of all the figures have been improved along with this the fond size has been increased.
- The Tables should be checked for consistent formats and expression errors, including units, spaces between values and units, etc.
We are apologetic for our recklessness. Now all the tabular columns have been revised.
- In the subsections, the authors are suggested to add some composite figures of typical research examples contained in this review, which can be helpful for better demonstration.
According the comment a composite figure has been added into the manuscript.
- The references should be checked with formats, and many errors exist such as ref. 103, 113, 123, etc. The manuscript should be checked and edited thoroughly for English. For example, expression errors of chemical formula in line 572, and English tense, etc.
As per the suggestion errors have been removed from the references. Along with this the expression errors have also been removed from the chemical formula. The manuscript has been checked thoroughly for any grammatical errors.
- The structures of this review needs to be improved to be clear considering that seven sections are contained. The authors discuss biosynthesis of various metal NPs yet the related contents are brief and limited.
According to the comment the manuscript has been revised thoroughly for seven sections and subsections. Along with this copper nanoparticle subheading has been added under the heading 'Synthesis of metal nanoparticles by entomopathogenic fungi' to increase the weightage.
Reviewer 2 Report
The manuscript entitled "Entomopathogenic Fungi: An eco-friendly synthesis of sustainable nanoparticles and their nanopesticide properties" reviews action of silver nanoparticles against pests.
Overall, this manuscript presents a valuable review of EPF-nanosilver particles and addresses entomopathogenic fungi in controlling the pest population. However, before publication in MDPI microorganism, I would like to emphasize the need for extensive editing in both scientific and English aspects.
Scientifically, the manuscript would benefit from further development in certain areas. For instance,
1. Little more attention can be given to highlighting the efficacy of biosynthesized nanoparticles as mentioned on page 3; line: 111.
I noticed that the summarization of Table: 1 lacks information regarding cytotoxicity and mechanism (either intra or extracellular), which is an essential aspect to consider when evaluating the potential applications of nanoparticles. For reference in toxicity Assessment of Silver Nanoparticles, I suggest you visit some review articles "Pantidos, Nikolaos, and Louise E. Horsfall. "Biological synthesis of metallic nanoparticles by bacteria, fungi and plants." Journal of Nanomedicine & Nanotechnology 5.5 (2014): 1."
2. A disadvantage of biological synthesis is the possibility of a difference in the concentrations of the biomolecules present in the biological source (whether microorganisms or plants) due to variations in the environment or the availability of nutrients. Please address how to overcome this issue with bio-nanoparticle synthesis.
3. A most familiar fungus like Aspergillus fumigatus, Fusarium oxysporum has been used in a large number of studies attempting to create metallic nanoparticles, especially those made of silver. I suggest you add these fungi to the table:1 and compare them to other fungi in the preparation of nanoparticles.
4. Please ensure that the data fed into table: 1 and 2 are summarized (for example, Table:1 shows the list of entomopathogenic fungi used in the preparation of Ag nanoparticles with its experimental parameters, but lack of summarization on the author's view upon which is preferred and what show low cell toxicity. Same with the case Table:2, lack of summarization in the author's view) to enhance the clarity and accuracy of the review article.
5. Although the extracellular mechanism for the biosynthesis of nanoparticles is preferred over the intracellular, a few advantages of intracellular mechanisms can be highlighted in this review manuscript. For reference, I suggest you visit some review articles "Pantidos, Nikolaos, and Louise E. Horsfall. "Biological synthesis of metallic nanoparticles by bacteria, fungi, and plants." Journal of Nanomedicine & Nanotechnology 5.5 (2014): 1."
6. In the line: 134, the Categorization of fungi with major 6 Phylum is depicted and the reason for choosing entomopathogens among them are not convincible, More reasons with proper citations would be appreciable.
7. Representations and labels in Figure: 2 are not clear, and need improvement.
Additionally, some key experiments or analyses may need to be included to strengthen the conclusions.
In terms of English, the manuscript requires thorough language editing to improve clarity and readability. Several grammatical errors need to be addressed.
Author Response
Manuscript Title : Entomopathogenic Fungi: An eco-friendly synthesis of sustainable nanoparticles and their nanopesticide properties
Reference No. : microorganisms-2410859
Reviewers comments
Reviewer # 2
- Little more attention can be given to highlighting the efficacy of biosynthesized nanoparticles as mentioned on page 3; line: 111.
I noticed that the summarization of Table: 1 lacks information regarding cytotoxicity and mechanism (either intra or extracellular), which is an essential aspect to consider when evaluating the potential applications of nanoparticles. For reference in toxicity Assessment of Silver Nanoparticles, I suggest you visit some review articles "Pantidos, Nikolaos, and Louise E. Horsfall. "Biological synthesis of metallic nanoparticles by bacteria, fungi and plants." Journal of Nanomedicine & Nanotechnology 5.5 (2014): 1.
We are thankful to the reviewer for pointing out our mistake. The efficacy of biosynthesized nanosilver has been added.
Additionally we have modified the Table 1 by adding intracellular and extracellular mechanism of synthesis. This table is regarding the synthesis of metal nanoparticles that is why cytotoxicity word could not be used, but the last sections of the manuscript provides a detailed description of the cytoxicity mechanism. We are extremely thankful for providing reference papers which helped in the correction of tabular column.
- A disadvantage of biological synthesis is the possibility of a difference in the concentrations of the biomolecules present in the biological source (whether microorganisms or plants) due to variations in the environment or the availability of nutrients. Please address how to overcome this issue with bio-nanoparticle synthesis.
We are very much obliged for this valuable suggestion given by the reviewer. We have already incorporated these points in the manuscript. (Line no:413).
- A most familiar fungus like Aspergillus fumigatus, Fusarium oxysporum has been used in a large number of studies attempting to create metallic nanoparticles, especially those made of silver. I suggest you add these fungi to the table:1 and compare them to other fungi in the preparation of nanoparticles.
As per the suggestion of the reviewer studies related to Aspergillus fumigatus and Fusarium oxysporum has been added to the tabular column.
- Please ensure that the data fed into table: 1 and 2 are summarized (for example, Table:1 shows the list of entomopathogenic fungi used in the preparation of Ag nanoparticles with its experimental parameters, but lack of summarization on the author's view upon which is preferred and what show low cell toxicity. Same with the case Table:2, lack of summarization in the author's view) to enhance the clarity and accuracy of the review article.
Thank you for the suggestion. We have already mentioned this details regarding the table in text of manuscript.
- Although the extracellular mechanism for the biosynthesis of nanoparticles is preferred over the intracellular, a few advantages of intracellular mechanisms can be highlighted in this review manuscript. For reference, I suggest you visit some review articles "Pantidos, Nikolaos, and Louise E. Horsfall. "Biological synthesis of metallic nanoparticles by bacteria, fungi, and plants." Journal of Nanomedicine & Nanotechnology 5.5 (2014): 1."
We are grateful for the comment, but after going through the above suggested paper it was seen that the advantages of intracellular synthesis is related to the bioremediation of heavy metals. That is the reason advantages of intracellular synthesis has not been incorporated in the manuscript.
- In the line: 134, the Categorization of fungi with major 6 Phylum is depicted and the reason for choosing entomopathogens among them are not convincible, More reasons with proper citations would be appreciable.
According to the suggestion we have added few studies along with their citations that can help in understanding the reason for choosing entomopathogens.
7. Representations and labels in Figure: 2 are not clear, and need improvement.
As per the suggestion we have increased the clarity and font size of figure 2
In terms of English, the manuscript requires thorough language editing to improve clarity and readability. Several grammatical errors need to be addressed.
As per the suggestion the manuscript has been edited and grammatical errors have been removed
Round 2
Reviewer 1 Report
I have checked all the revisions. The revised manuscript has improved some of the contents, yet the following problems still need to be addressed:
1. For the downloaded PDF, Figures were really hard to read, such as the texts in Figure 1 and 3. Please have a check.
2. In the Introduction, the references should be enriched to better show advantages and recent progress of biosynthesis including 10.1002/sstr.202200356, which is suggested to be useful.
3. Three Figures are quite simple. In the subsections, the authors are suggested to add one/two more composite figures of typical research examples, which can be much helpful for better demonstration.
4. The format of references should be checked thoroughly. For example, citing [7, 8, 9, 10, 11, 12, 13, 14] in the body and [7-14] maybe correct.
Author Response
Manuscript ID: microorganisms-2410859
Reviewer 1
- For the downloaded PDF, Figures were really hard to read, such as the texts in Figure 1 and 3. Please have a check.
We fully agree with the suggestion. We have tried our best to enhance the Figures
- In the Introduction, the references should be enriched to better show advantages and recent progress of biosynthesis including 10.1002/sstr.202200356, which is suggested to be useful.
As per the suggestion this paper has been added in the introduction part
- Three Figures are quite simple. In the subsections, the authors are suggested to add one/two more composite figures of typical research examples, which can be much helpful for better demonstration.
We are thankful for the suggestion, but in the last revision we have already incorporated a composite figure.
- The format of references should be checked thoroughly. For example, citing [7, 8, 9, 10, 11, 12, 13, 14] in the body and [7-14] maybe correct.
According to the comment citation format has been changed in the body
